# The Role of Macrophages in Sarcoma Tumor Microenvironment and Treatment

**DOI:** 10.3390/cancers15215294

**Published:** 2023-11-05

**Authors:** Agnieszka E. Zając, Anna M. Czarnecka, Piotr Rutkowski

**Affiliations:** 1Department of Soft Tissue/Bone Sarcoma and Melanoma, Maria Sklodowska-Curie National Research Institute of Oncology, 02-781 Warsaw, Poland; agnieszka.zajac@nio.gov.pl (A.E.Z.); piotr.rutkowski@nio.gov.pl (P.R.); 2Department of Experimental Pharmacology, Mossakowski Medical Research Centre, Polish Academy of Sciences, 02-176 Warsaw, Poland

**Keywords:** tumor immunology, macrophages, tumor-associated macrophages, bone sarcomas, soft tissue sarcomas

## Abstract

**Simple Summary:**

Soft tissue and bone sarcomas belong to a group of rare and malignant tumors. The treatment of these tumors is very complex and depends on their specific subtypes. Therefore, there is a great need to develop novel therapeutic options for patients with sarcomas. One such option may be to target tumor-associated macrophages (TAMs), which are involved in immunosuppression during tumor growth. High levels of TAMs are widely recognized as a poor prognostic factor in many tumors, including sarcomas, making them a promising target for future targeted therapies. In this review, we highlight the role of TAMs in the microenvironment of sarcomas, along with their clinical relevance, potential targetable markers on their surface, and the molecular pathways involved. We also discuss currently ongoing clinical trials with TAMs.

**Abstract:**

Sarcomas are a heterogeneous group of malignant mesenchymal tumors, including soft tissue and bone sarcomas. Macrophages in the tumor microenvironment, involved in immunosuppression and leading to tumor development, are called tumor-associated macrophages (TAMs). TAMs are very important in modulating the microenvironment of sarcomas by expressing specific markers and secreting factors that influence immune and tumor cells. They are involved in many signaling pathways, such as p-STAT3/p-Erk1/2, PI3K/Akt, JAK/MAPK, and JAK/STAT3. TAMs also significantly impact the clinical outcomes of patients suffering from sarcomas and are mainly related to poor overall survival rates among bone and soft tissue sarcomas, for example, chondrosarcoma, osteosarcoma, liposarcoma, synovial sarcoma, and undifferentiated pleomorphic sarcoma. This review summarizes the current knowledge on TAMs in sarcomas, focusing on specific markers on sarcoma cells, cell–cell interactions, and the possibly involved molecular pathways. Furthermore, we discuss the clinical significance of macrophages in sarcomas as a potential target for new therapies, presenting clinical relevance, possible new treatment options, and ongoing clinical trials using TAMs in sarcoma treatment.

## 1. Introduction

Macrophages can be divided into two main groups, called classically activated macrophages (M1) and alternatively activated macrophages (M2). Both M1 and M2 macrophages have distinct functions in immune defense and surveillance and can be interconverted by changes in the internal environment [1]. Macrophages that are recruited to the tumor microenvironment (TME) are called tumor-associated macrophages (TAMs) [2]. TAMs do not fit well in the M1 or M2 group, because they share the characteristics of both M1 and M2 groups, and they present another subpopulation of macrophages [3,4,5]. TAMs play a particular role in the regulation of the TME, leading to immunosuppression, tumor growth, and migration in many tumors, including sarcomas [2,6].

Sarcomas are a group of various malignant mesenchymal tumors that has more than 100 different subtypes, including soft tissue sarcomas (STS) and primary bone sarcomas [7,8]. The treatment of sarcomas is challenging and complex and depends on the specific subtypes of sarcoma [2]. High levels of TAMs are generally recognized as a weak prognostic factor in various types of cancers, making them a potential target for future targeted therapies in sarcoma [9,10]. 

This review highlights the role of TAMs in STS and bone sarcoma TME, discussing the specific markers and cell interactions involved in tumor development among different types and subtypes of sarcomas. Furthermore, we describe the clinical significance of macrophages, with their role as future therapeutic targets in sarcomas.

## 2. M1 and M2 Macrophages

### 2.1. M1 and M2 Characteristics

M1 macrophages play a crucial role in the functioning of the human immune system and may participate in tissue destruction. They have pro-inflammatory and antitumor characteristics, with strong lethal effects on pathogens entering the body, in contrast to M2 macrophages [11]. M1 macrophages can capture, phagocytize, and lyse antigen-presenting tumor cells, promoting the cytotoxic function of other immune cells, such as CD8+ T cells and NK cells [12]. M1 macrophages are characterized by the secretion of cytokines, which modify the immune microenvironment, such as tumor necrosis factor α (TNF-α), IFN-γ, and interleukins: IL-1β, α, IL-12, IL-6, IL-18, chemokine (C-X-C motif) ligand (CXCL) 10, and CXCL9 [13,14]. M1 macrophages express high levels of the main histocompatibility complex class II (MHC-II) cell surface receptor (HLA-DR), C-C Motif Chemokine Receptor 7 (CCR7, CD197), CD68, CD40, CD11c, and the costimulatory molecules CD80 or CD86 [11,12,13,15]. M1 macrophages also produce high levels of reactive oxygen species (ROS) and nitric oxide synthase (NOS), an enzyme that metabolizes arginine to NO [6,11] (Table 1). 

M2 macrophages are involved in parasite infection, tissue repair, and remodeling through immune tolerance, wound healing, allergic diseases, and the regulation of angiogenesis [11]. Unlike M1 macrophages, M2 macrophages secrete anti-inflammatory and pro-tumoral cytokines and are polarized by Th2-derived cytokines [11]. M2 macrophages release many growth factors, such as TGF-β, platelet-derived growth factor (PDGF), hepatocyte growth factor (HGF), and basic fibroblast growth factor (bFGF) [12]. Furthermore, through the secretion of adrenomedullin and vascular epithelial growth factors (VEGFs), and the expression of several immunosuppressive molecules (such as IL-10, programmed death-ligand 1 (PD-L1), or TGF-β, promoting tumor growth), M2 macrophages support angiogenesis in cancer cells [13,16]. Mantovani et al. [17], additionally divided M2 macrophages into four subtypes: M2a (induced by fungal and helminth infections, IL-4 and IL-13), M2b (induced by an immune complex and LPS), M2c (induced by IL-10, TGF- β, and glucocorticoids (GC)), and M2d (induced by IL-6 and adenosine) [17,18]. Each of these subtypes may present specific markers on their surface, which are presented in Table 1.

**Table 1 cancers-15-05294-t001:** Subset phenotypes of classically activated macrophages (M1) and alternatively activated macrophages (M2).

	M1	M2a	M2b	M2c	M2d	Reference
Stimulation/ activation	IFN-γ LPS GM-CSF	IL-4 IL-13 Fungal and Helminth infection M-CSF	Immune complexes IL-1R	IL-10 TGF-β GCs	IL-6 Adenosine	[3,8,11,12,13,17]
Marker expression	CD11c CD1a CD16 CD32 CD40 CD54 CD68 CD86 CD80 CD197 MHC-II IL-1R IL2RA TLR2 TLR4 iNOS SOCS3	Arg1 CD36 CD163 * CD200R CD204 CD209 (DC-SIGN) CSF1R MHC-II SR-A MMR (CD206) MGL1/2 TGM2 DcR3 IL-1R II	Arg1 CD80 CD86 MHC-II	Arg1 CCR2 CD36 CD163 TLR1 TLR8 SLAM	VEGF	[11,12,13,15,17,18,19,20,21,22]
Cytokine secretion	TNF-α IL-1 β IL-6 IL-12 IL-23	IL-10 TGF-β IL-1ra	IL-1 IL-6 IL-10 TNF-α	IL-10 TGF-β	IL-10 IL-12 TNF-α TGF-β	[8,13,16,17]
Chemokine secretion	CCL2 CCL3 CCL4 CCL5 CCL8 CCL9 CXCL8 CXCL9 CXCL10 CXCL11 CXCL16 CCR7	CCL2 CCL17 CCL18 CCL22 CCL24	CCL1	CCL13 CCL16 CCL18	CCL5 CXCL10 CXCL16	[12,14,17,18]

* The marker CD163 is not unique to M2 cells, as it is related to the Avian Musculoaponeurotic Fibrosarcoma Oncogene transcription factor V-Maf (c-MAF) [23]. Arg1—arginase 1, CCL—chemokine ligand of the C-C motif, CCR—chemokine receptor of the C-C motif chemokine, Chil3/4—chitinase-like protein Ym1/2, CSF—colony-stimulating factor, CSF1R—colony-stimulating factor receptor 1, CXCL—chemokine ligand (C-X-C motif), DcR3—Decoy receptor 3, DC-SIGN—dendritic cell-specific intercellular adhesion molecule-3-Grabbing non-integrin, FIZZ1—resistin-like molecule α, GC—glucocorticoids, GM-CSF—colony-stimulating factor, IL2RA—interleukin 2 receptor subunit alpha, IFN—interferon, iNOS—inducible nitric oxide synthase, LPS—lipopolysaccharide, MHC—main histocompatibility complex, MGL—macrophage galactose-type lectin, MMR—macrophage mannose receptor C-type, SLAM—signaling lymphocytic activation molecule, SOCS—suppressor of cytokine signaling, SR-A—macrophage scavenger receptor A, TGF—transforming growth factor, TGM—transglutaminase, TLR—Toll-like receptor, TNF—tumor necrosis factor, VEGF—vascular epithelial growth factor.

### 2.2. Macrophage Polarization

Based on different stimuli, macrophages can polarize into M1 or M2 macrophages with different activation ways. Classical macrophage activation, which occurs in M1 macrophages, is a result of cell stimulation by many factors, including (1) interferon-gamma (IFN-γ), secreted primarily by Th1 helper CD4+ T cells, cytotoxic CD8+ T cells, and natural killer cells (NK); (2) lipoproteins and lipopolysaccharide (LPS), an external membrane component of Gram-negative bacteria; (3) granulocyte-macrophage colony-stimulating factor (GM-CSF, CSF2), which stimulates the production of inflammatory cytokines. Bacterial LPS and lipoproteins stimulate the Toll-like receptor (TLR) ligands TLR4 and TLR2, which activate interferon regulatory factors (IRFs)—IRF3, IRF5 and IRF7—signal transducer and activator of transcription 1 (STAT1), and P50-P65 nuclear factor-κB (NFκβ) gene, which finally lead to M1-associated gene activation [13]. On the other hand, the alternative activation of macrophages is stimulated by macrophage colony-stimulating factor (M-CSF1, CSF1), IL-4, IL-10, transforming growth factor β (TGF-β), IL-13, and fungi or helminth infections, and leads to the M2 phenotype [14]. Another important factor in macrophage polarization is hypoxia, which induces the phenotype of M1 or M2 macrophages, respectively, by hypoxia-inducible factor (HIF)1α or HIF2α [13].

As mentioned before, CSF2 and IFN-γ induce the expression of IRF5, which activates M1 macrophages and inhibits M2 macrophages. Furthermore, CSF2R participates in the repolarization of M2 macrophages towards an M1 phenotype [13]. The upregulation of the suppressor of cytokine signalling (SOCS) 3 suppressor through STAT1/STAT2 is also crucial for the efficient activation of M1 macrophages [14]. A study by Arnold et al. [24] showed that SOCS3 expression regulates phosphatidylinositol 3-kinase (PI3K) activity, favors nitrogen (II) oxide (NO) production, and leads to the activation and translocation of Nuclear-kappa B factor (NF-κB). Furthermore, activin A has been observed to promote the expression of the typical markers of M1 macrophages by negatively regulating IL-10 production, likely leading to M1 polarization [14].

On the other hand, CSF1, TGF-β, interleukins: 4, 6, 13, 10, and prostaglandin E2 (PGE2), released by tumor cells, can cause macrophage polarization into M2 macrophages [1,11,14]. Other molecules, involved in M2 polarization, are IRF4, which is a negative controller of TLR signaling associated with the MyD88 adaptor molecule, or NFκB [14]. Furthermore, Ohmori et al. [25] showed that IL-4 inhibits IFN-γ-stimulated gene transcription and increases STAT6 activation, affecting arginase 1(Arg1) production by M2 macrophages. On the other hand, IL-21 influences polarization in M2 macrophages by inhibiting extracellular signal-regulated kinase (ERK) phosphorylation and reducing inducible nitric oxide synthase (iNOS), which in turn increases STAT3 phosphorylation [14]. Furthermore, the complement 32 response gene (RGC-32), which is a cell cycle regulator, also plays an important role in the polarization of M2 macrophages [8]. The factors presented influence the phenotype of macrophages, which varies depending on the stimuli they receive. Some of these factors are present in TME and are involved in the development of specific macrophages that participate, e.g., in tumor growth.

## 3. Tumor-Associated Macrophages (TAMs)

### 3.1. Characteristics of TAMs

Macrophages are relocated to tumor areas and differentiate into TAMs in response to various cytokines, such as IL-34, CSF1, members of VEGFs, and chemokines, such as CCL5 and CCL2 [2]. The TME influences the secretion of cytokines, chemokines, and other agents released by mesenchymal cells, tumor cells, or immune cells. These factors, in the presence of inflammation, local hypoxia, and high levels of lactic acid, cause the relocation of monocytes from blood into the TME and repolarize them into a particular type of macrophage, called a TAM. TAMs are more polarized towards M2 macrophages, which refers to more aggressive tumor hallmarks, represented by tumor invasion, progression, and metastasis [11]. The polarization of TAMs into the pro-tumor phenotype can also be caused by dying cancer cells. For example, apoptotic cells release sphingosine-1-phosphate (S1P), which attracts and activates TAMs by reducing IL-12 production and downregulating MHC-II expression. Necrotic cells, on the other hand, can promote tumor growth by recruiting TAMs to the TME and releasing IL-1α [20].

Although TAMs are more similar to M2 macrophages, they have both M1 and M2 polarity signatures [14]. In response to different stimulants, TAMs can differentiate into separate functional subgroups. Th1 T cells (by secreting IFN-γ, TNF-α, LPS, etc.) lead TAMs to transform into M1 macrophages, and Th2 T cells (by secreting TGFβ1, IL-10, IL-4, PGE2, etc.) transform them into M2 macrophages [12]. Therefore, TAMs express both M1 and M2 markers, such as MGL1, CD81, MHC-II, SR-A, PD-L1, CD163, MRC1, CD206, CD204 etc. [2,14,26].

### 3.2. TAMs Function in Tumor Development

TAMs influence both the TME and tumor cells (Figure 1). In the TME, TAMs secrete various cytokines, chemokines, and enzymes, such as, e.g., TGF-β, prostaglandins, IL-10, CCL22, CCL17, galectin-3 (Gal-3), and metalloproteinases (MMP), that induce immunosuppression by the activation of regulatory T cells (TREG), which, in consequence, lead to the loss of T cell function to recognize and kill tumor cells [1,26,27]. TAMs can inhibit cytotoxic CD8+ T cells by the metabolic starvation of T cells through the secretion of immunosuppressive metabolites (i.e., NO, ROS) [28,29]. Many other suppressive markers, interacting with tumor cells, are observed in TAM, such as, for example, (1) programmed death cell receptor 1 (PD-1), a PD-L1 ligand in tumor cells; (2) CD80, a CD28/cytotoxic T lymphocyte Associated Protein 4 (CTLA-4) ligand; (3) V-domain Ig suppressor of T cell activation (VISTA); (4) signal regulatory protein alpha (SIRPA), a ligand for the CD47 receptor; (5) leukocyte immunoglobulin-like receptor subfamily B member 1 (LILRB1), an MHC-I ligand; or (6) sialic acid-binding Ig-like lectin 10 (Siglec-10), a CD24 ligand [1,2]. The PD-1/PD-L1 signalling pathway enhances tumor immune escape by inhibiting macrophage normal function and binding Siglec-10 to CD24. In addition, it limits the functions of T effector cells, NK cells, and dendritic cells (including the suppression of activation, proliferation, and cytokine expression) [30]. On the other hand, The SIRPA/CD47 pathway is defined as the signal ‘do not eat me’, which recognizes CD47-expressing tumor cells as self-normal cells and results in the inhibition of phagocytosis. LILRB1, located on the surface of macrophages, is recognized by MHC-I molecules and changes macrophage function from tumor growth inhibition to tumor promotion [1]. In addition, vascular cell adhesion molecule 1 (VCAM-1, CD106), which mediates TAM adhesion to the vascular endothelium, is expressed in TAMs [14].

In addition to the modulation of the TME, TAMs promote tumor development and progression by releasing factors that enhance tumor growth, invasion, angiogenesis, and metastasis potential [29]. Many cytokines released by TAMs are involved in tumor growth. For example, through IL-6 secretion, TAMs increase the capacity for carcinogenesis and tumor progression by controlling genes related to the cell cycle, angiogenesis, inflammation, and stem cell self-renewal promotion [1,10]. IL-6 participates in the STAT3 signaling pathway, promoting tumorigenesis [10] and the induction of cancer stem cells (CSCs) [31]. In tumor development are also involved other factors secreted by TAMs, eg IL-10, that support tumor growth and aggressiveness through the PI3K signaling pathway and the Janus-Activated Kinase (JAK)1/STAT1/NF-kB/neurogenic locus homolog protein 1 (Notch1) pathway [1,10], or CCL2, which activates the PI3K/protein kinase B (Akt)/mammalian target of rapamycin (mTOR) [10]. Furthermore, the CSF2 receptor (CSF2R) in TAMs is involved in the regulation of JAK2/STAT3/5, mitogen-activated protein kinase (MAPK), NFκB, and PI3K signaling pathways [13].

In addition, TAMs also support angiogenesis. In these processes, the key roles are of VEGFs in promoting the vascularization process, and MMPs in promoting the degradation of the extracellular matrix (ECM) [10,29]. Furthermore, tumor angiogenesis is encouraged by the IL-1β, IL-8, TNF-α, PDGF, and basic bFGF, secreted by TAMs [2]. VEGFs and MMPs are also involved in tumor invasion. This process is related to the epithelial–mesenchymal transition (EMT), which can be mediated by the secretion of IL-6 and TGF-β, through the JAK/STAT3/Snail and Smad/Snail pathways, respectively [15,29]. EMT can also be induced by epidermal growth factor (EGF) secretion, activating the epidermal growth factor receptor (EGFR)-ERK signaling pathway [10]. MMPs, EGF, and CSF-1 factors enhance cell migration and are involved in metastasis [29]. Furthermore, tumor metastasis is supported by IL-8 secretion through the upregulation of engulfment and cell motility 1 (ELMO1) expression, a tumor-expressed protein involved in cell migration, phagocytosis, and morphological changes [1].

## 4. TAMs in Sarcomas

### 4.1. TAMs Phenotype and Role in Sarcoma

TAMs have been widely identified among sarcomas. They have been found in STS (in 90%, in a study of various types of sarcomas, including 14 gastrointestinal stromal tumors (GISTs), five synovial sarcomas (SSs), four leiomyosarcomas (LMSs), and one–three cases of 18 other types) [32]. In addition, they have also been identified in bone sarcomas (e.g., osteosarcoma, Ewing sarcoma (ES), and chondrosarcoma) [2]. The TAMs presented in STS and bone sarcomas have various markers identified on their surface (Figure 2).

One of these markers is SIRPA [33], which is a glycoprotein of the SIRP family, located mainly on the membrane of myeloid cells [34], but it is also identified on the surface of macrophages, including TAMs [2,13,35]. It has been identified in both STS and bone sarcoma [33]. In macrophages, SIRPA recognizes CD47, which negatively regulates the effector function of innate immunity cells, such as the phagocytosis of the host, and participates in homeostasis processes by removing dying cells [36]. The inhibition of phagocytic potency against tumor cells is also caused by the expression of PD-1 in TAMs [37]. PD-1 expression has been observed in the human osteosarcoma mouse model and patient samples of osteosarcoma with lung metastases [38]. In this study, anti-PD1 antibodies led to a decrease in the number of tumor cells and enhanced tumor apoptosis by blocking the p-STAT3/p-Erk1/2 signaling pathway. Furthermore, an increased number of M1 macrophages and NK cells was observed after PD-1 blockade [38].

The following marker is the tyrosine kinase receptor CSF1R, expressed in circulating monocytes and macrophages, which is a ligand for CSF1 and IL-34. These factors are secreted by tumor cells and cause the recruitment of TAMs [2,39]. CSF1 is highly expressed in several types of solid tumors, and in addition to recruiting peripheral blood monocytes to the TME, it plays a role in the differentiation of monocytes into macrophages. Furthermore, CSF1 participates in polarization to an M2-like phenotype by binding to CSF1R [2]. In a study by Fujiwara et al., the in vitro administration of Pexidartinib (PLX3397) reduced TAM M2-like polarization to an M1-like phenotype with a simultaneous decrease in forkhead box P3 (FOXP3)+ regulatory T cells and the increased migration of CD8+ T cells [40]. Moreover, in the same study, in the osteosarcoma mice model, PLX3397 administration suppressed the primary tumor growth and lung metastasis in vivo [40]. On the other hand, IL-34 induces the adhesion of monocytes to endothelial cells, improving the migration capacity of TAMs, which was observed in osteosarcoma [41]. Sarcoma cells may also secrete the macrophage migration inhibitory factor (MIF) pleiotropic cytokine, which interacts with the CD74 receptor, expressed in macrophages, and translates their phenotype into pro-tumoral, as it was observed in in vitro and in vivo studies in the STS mouse model [42].

On the other hand, CCL22, IL-10, TGF-β, and indoleamine 2,3-dioxygenase (IDO) 1, secreted by TAMs, lead to immunosuppression, resulting in the inhibition of effector T cells and TREG activation [13,16]. One of them, IDO1, is a principal regulator of the tryptophan (Trp) catabolism pathway, leading to the loss of antitumor immunity and tumor growth [43]. Kynurenine, produced by IDO1, promotes the selective expansion of Tregs [44]. In this process, the PD-L1 and CD80 receptors are also involved, leading to T-cell dysfunction [45,46]. For example, in a preclinical model of osteosarcoma, the inhibition of the PD-1/PD-L1 interaction has been associated with antitumor activity [38].

Factors secreted by TAMs may also influence tumor growth in sarcomas, e.g., pro-inflammatory cytokines are specifically implicated in the pathogenesis of chondrosarcoma and osteosarcoma [39,47]. For example, increased TNF-α, MMP-12, and MMP-13 expression were observed in chondrosarcoma cells during malignant transformation [47,48]. However, MMP-9 increased the metastatic potential in osteosarcoma [39]. Another factor, TGF-β, has a key role in human chondrosarcoma migration and metastasis, for example, by increasing human chondrosarcoma mobility through the PI3K/Akt pathway [49]. TAMs may also promote invasiveness by secreting CCL18 and pro-inflammatory molecules, such as cyclooxygenase-2 (COX2), which were observed in metastatic osteosarcoma [39]. Furthermore, IL-6 promotes stemness in osteosarcoma by up-regulating the signaling pathway regulating EMT, through STAT3 phosphorylation. On the other hand, IL-34 has been reported to induce vasculogenesis and tumor growth in osteosarcoma through the activation of endothelial cell signaling pathways, including the proto-oncogene tyrosine protein kinase Src (Src), focal adhesion kinase (FAK), and MAPKs, involved in angiogenesis and vascular homeostasis [41]. Furthermore, in osteosarcoma, TAMs promoted angiogenesis through the secretion of VEGF [4,39].

Other markers, characteristic of sarcoma TAMs, are CD163 and CD68, specific for M2 and M1 macrophages, respectively [6]. They were commonly observed among sarcomas, for example, in chondrosarcoma and osteosarcoma or angiosarcoma, and dedifferentiated liposarcoma [33,35,50,51]. In a study by Burns et al., a poor correlation between CD163 and STAT6 was found in dedifferentiated liposarcoma and in undifferentiated pleomorphic sarcoma (UPS) [52]. A study by Handl et al. [51] of 24 patients with primary ES has shown that CD163 is a sensitive marker for macrophages in this group of sarcomas, although it lacks specificity for the M2 phenotype.

Other M2-like TAMs, identified in sarcomas, are, for example, mannose receptor-1 (MMR, CD206), observed in dermatofibrosarcoma protuberans (DFSP), and osteosarcoma, macrophage scavenger receptor 1 (MSR1, CD204), observed in UPS, and dendritic cell-specific ICAM-3 capture non-integrin (DC-SIGN, CD209) and melanoma cell adhesion molecule (MCAM, CD146), observed in osteosarcoma [2,39]. Moreover, in a study by Jiang et al., the expression of structural maintenance of chromosomes 4 (SMC4) was significantly correlated with CD14+ and CSF1R+ monocytes, CD80+ TAMs, prostaglandin-endoperoxide synthase 2 (PTGS2)+ M1 macrophages, and the expression of depleted CTLA4+ T cells [53]. This suggests that SMC4 may play a role in regulating the infiltration and activity of TAM and T cell function inhibition and, therefore, may promote the development of sarcoma [53].

In the study by Dancsok et al. [33], in a cohort of patients with approximately 960 cases of sarcomas (including STSs: DFSP, extraskeletal myxoid chondrosarcoma (EMC), UPS, GIST, low-grade fibromyxoid sarcoma (LGFMS), LMS, myxofibrosarcoma (MFS), malignant peripheral nerve sheath tumor (MPNST), epithelioid sarcoma, alveolar soft part sarcoma (ASPS), SS, angiosarcoma, liposarcoma, rhabdomyosarcoma, clear cell sarcoma; and bone sarcomas: chondrosarcoma, chordoma, osteosarcoma, ES), among almost all analyzed cases, TAMs outnumbered the tumor-infiltrating lymphocytes (TILs). The same observation was made in a study by Fan et al. [54] of 472 samples with eight different diagnosed sarcomas (including 123 osteosarcomas and 211 STS, such as ES, fibrosarcoma, LMS, liposarcoma, SS, UPS, etc.), where two of the most common fractions of immune cells were M2 macrophages (almost 34%) and nonactivated macrophages (M0) (around 21%). In another cohort, including 253 localized STSs and GISTs, among many immune cells, the highest reported levels were of monocytes/macrophages [55].

Several markers of TAMs have already been described among different subtypes of sarcomas. One such marker is CD168. The high infiltration (85%) of CD168+ TAMs was observed, e.g., in dedifferentiated chondrosarcoma, exclusively in dedifferentiated areas, and not in the well-differentiated component of chondrosarcoma [56]. In a study by Richer et al. [35], TAMs appeared to be the most abundant immune population both in the peritumoral area of conventional chondrosarcomas and in the dedifferentiated areas of dedifferentiated chondrosarcomas, compared to TILs (25% were CD163+ TAMs in conventional and 50% in dedifferentiated, compared to, respectively, 10% and 15% being CD3+ cells, and 8% and 5% being CD8+ cells). CD163+ macrophages were also identified in another cohort of 26 grade 1–3 conventional chondrosarcomas and were localized mainly at the peripheral site of the tumor [57]. On the other hand, in osteosarcoma, CD14+HLA-DR+ M1 macrophages and CD14+CD163+ M2 macrophages were observed [58]. Furthermore, in another study, a heterogenic population of TAMs with the phenotype of both M1 and M2 macrophages in osteosarcomas was identified [39]. M2-like TAMs induced resistance to chemotherapy in primary osteosarcoma cells, by releasing IL-1β, which in response caused the overexpression of IL-34, vasculogenesis, and osteosarcoma growth [39].

However, the CD163 marker has also been identified in STS. In a study of 134 adults with high-grade LMS, liposarcomas, and SS, a high number of CD163+ TAMs was identified in 49% of all cases and were correlated with the tumor grade in 53% of high-grade and 28% of low-grade tumors [59]. However, the CD168+ TAM distribution was higher in LMS (66%), compared to liposarcoma and SS (respectively, 46% and 9%) [59]. The researchers also observed that STSs and bone sarcomas have a higher number of M2-like TAMs (marked with the CD163 marker) than M1-like TAMs. The M2: (M0+ M1) ratio was also higher in sarcomas without translocations, although SS was prone to have more M0 than M2 macrophages [33]. Furthermore, GISTs, LMSs, and UPSs were noticeably infiltrated by CD163+ macrophages expressing IDO1 [60].

Another marker, CD68, was particularly prominent among translocation-free sarcomas, especially pleomorphic liposarcoma, chordoma, chondrosarcoma (conventional and dedifferentiated subtypes), UPS, and angiosarcoma [33,61]. These sarcomas had a significantly higher CD68: TIL ratio than translocation-associated sarcomas, which was confirmed by immunohistochemistry and the analysis of mRNA expression [33]. Furthermore, in a validated cohort of 27 conventional and 49 dedifferentiated chondrosarcomas, a low expression of CD68+ TAMs was identified in approximately 39% of the patients, while its high expression was found in almost 61% [35].

The other TAM marker, SIRPA, was identified in macrophages in 31% of the sarcomas, especially chordoma (71% of the cases), dedifferentiated liposarcoma (77%), angiosarcoma (75%), and well-differentiated liposarcoma (65%) [33]. SIRPA+ TAM infiltration in osteosarcoma and chondrosarcoma was less common (respectively, observed in around 30% and 2% of all cases). SIRPA+ TAMs were observed in both conventional and dedifferentiated chondrosarcoma [35]. In all of these sarcomas, the expression of the SIRPA receptor was poorly correlated with the expression of CD47 in tumors. Furthermore, the expression of CD47 and SIRPA showed poor positive correlations with CD68+ and CD163+ macrophages in almost all sarcomas, except a solitary fibrous tumor, demonstrating a significant negative correlation between CD163+ and CD68+ TAMs, and CD47 [33]. SIRPA was also positively correlated with PD-L1 expression in TAMs in osteosarcoma, myxofibrosarcoma, dedifferentiated liposarcoma, and SS [33].

On the other hand, CSF1R expression was observed in STSs with complex genomic profiles and chondrosarcoma [35,55]. In chondrosarcoma, the expression of CSF1R was involved in macrophage survival and proliferation and varied between conventional and dedifferentiated chondrosarcoma (64% of the conventional chondrosarcomas, and around 85% of the dedifferentiated chondrosarcomas were positive for CSF1R). CSF1R+ macrophages were found in the peripheral areas of grade 2 and 3 chondrosarcomas and in the dedifferentiated compartment of dedifferentiated chondrosarcomas [35].

In addition to the TAM markers mentioned above, in a study by Han et al. [62], patients with aggressive pediatric bone sarcomas showed an increased number of peripheral CD14+ HLA-DR low/immunosuppressive monocytes and CTLA-4+ T cells, as well as CD14+ macrophage infiltrates. The presence of CD14+CD68+ M1 macrophages was also observed in a study of primary malignant bone tumors [63]. Furthermore, in a cohort of 253 localized STSs and GISTs, the TAMs also expressed CD16a as well as CSF1R, CD163, and CD68 [55]. Another less common marker, described in TAMs, was the ionized calcium-binding adaptor molecule 1 (Iba1), observed in UPS [64].

The distribution of TAMs among sarcomas is very heterogeneous, as a group of tumors. For this reason, there is still not much known about the specific markers of TAMs and their interaction in sarcomas. This may impact the clinical relevance of TAMs and their conflicting function on clinical outcomes among different subtypes of sarcomas.

### 4.2. The Role of TAMs as a Predictive and Prognostic Factors in Sarcoma

The high infiltration of TAMs in TME is generally associated with a poor overall survival (OS) rate and metastasis in STS and bone sarcomas [2,35,39] (Table 2). For example, a high infiltration of CD68+ TAMs tended to be associated with worse overall survival in the dedifferentiated chondrosarcoma with the osteosarcoma compartment, and the CD68/CD8 ratio was identified as an independent weak prognostic factor of OS in these patients [35]. Furthermore, all patients with metastasis at diagnosis had higher levels of CD68+ and CD163+ TAMs [35]. CD68+ TAMs were also an adverse prognostic factor that predicted a lower OS in ES, and worse progression-free survival (PFS) in LMS [33,51]. Furthermore, higher levels of CD68+ TAMs were associated with a poorer OS in myxoid liposarcoma that promoted invasiveness through the heparin-binding EGF-like growth factor (HB-EGF)-EGFR-PI3K/Akt pathway [65]. A study of almost 200 STSs indicated that CD68+ macrophages were independently associated with an increased risk of LR, regardless of margin status, age, sex, or the number of CD20+ B cells [61]. CD68+ and CD163+ TAMs were related to worse disease-specific survival (DSS) in non-gynecologic LMS, while they were not associated with DSS in gynecologic LMS [66].

Furthermore, the highest CD163+ M2 macrophage density was found in the highest grade of chondrosarcoma, suggesting the involvement of M2 macrophages in tumor progression through its role in angiogenesis, migration, or invasion [35,67]. In a study by Dancsok et al. [33], solitary fibrous tumors with CD163+ macrophages had a worse prognosis (with lower PFS). Other studies revealed that CD14+, CD68+, and CD163+ macrophages were associated with a poor OS in myxoid liposarcoma and dedifferentiated liposarcoma [68,69]. CD163+ macrophages were also associated with a worse OS and PFS in SS and with a poor OS in UPS [70,71]. Furthermore, in a study by Schroeder et al. [72], of 147 patients with sarcoma treated with trabectedin (including UPS, LMS, well-/dedifferentiated liposarcoma, SS, and uterine LMS), the gene signatures of myeloid-derived suppressor cells and M2 macrophages, and the high expression of M2 macrophages were significantly associated with a high death risk in the 7-year OS. In a study of 75 STS cases, a high number of CD68+, CD163+, and CD204+ macrophages in both the intratumoral and marginal areas was associated with worse disease-free survival (DFS); however, after the division of this group into grade 1 and grade 2/3, only in the grade 2/3 group were high numbers of CD163+ and CD204+ macrophages associated with poor DFS, and only in the marginal area [73].

Similarly, SIRPA+ and CSF1R+ TAMs were related to the metastatic status at diagnosis and a poor OS in patients with dedifferentiated chondrosarcoma [35]. Furthermore, the appearance of SIRPα+ macrophages was also an adverse prognostic agent in both SS and myxofibrosarcoma, and it was associated with a worse OS in sarcomas without translocations [33]. In UPS, in addition to CD163+ M2 macrophages, CD204+ M2 macrophages were associated with a worse OS [64]. Among other markers, in osteosarcoma, TAMs expressing CD47 predicted an inferior PFS [33].

However, TAMs can have also a positive prognostic role in some cases (Table 2). A study by Handl et al. [51], has shown an association between the CD163 expression of TAMs and a lower stage of ES. A higher presence of CD163+ macrophages was associated with localized disease and a longer OS [2,51]. The same results were obtained in osteosarcoma, where CD163+ macrophages were significantly correlated with CD146+ and macrophage activation factor (cMAF)+ macrophages and were associated with a higher OS rate [2,50,74]. In dedifferentiated liposarcoma and LMS, a high infiltration of CD163+ TAMs was a prognostic factor with a longer PFS [33].

**Table 2 cancers-15-05294-t002:** Summary of various tumor-associated macrophages (TAMs) and their clinical relevance among sarcomas.

Tams’ Marker	Sarcoma Subtype	Overall Survival Rate	Progression-Free Survival	Metastatic Potential	Reference
CD68	dedifferentiated chondrosarcoma	decreased	-	increased	[35]
	myxoid liposarcoma	decreased	-	-	[65]
	dedifferentiated liposarcoma	decreased	-	-	[69]
	Ewing sarcoma	decreased	-	-	[50]
	leiomyosarcoma	-	decreased	-	[33]
CD163	synovial sarcoma	decreased	decreased	-	[71]
	undifferentiated pleomorphic sarcoma	decreased	-	-	[70]
	solitary fibrous tumors	-	decreased	-	[33]
	myxoid liposarcoma	decreased	-	-	[68]
	dedifferentiated liposarcoma	decreased	increased	-	[33,69]
	osteosarcoma	increased	-	increased	[50,75]
	Ewing sarcoma	increased	-	-	[51]
	leiomyosarcoma	-	increased	-	[33]
SIRPA	dedifferentiated chondrosarcoma	decreased	-	increased	[35]
	synovial sarcoma	decreased	-	-	[33]
	myxofibrosarcoma	decreased	-	-	[33]
CSF1R	dedifferentiated chondrosarcoma	decreased	-	increased	[35]
CD14	osteosarcoma	increased	-	-	[58]
	myxoid liposarcoma	decreased	-	-	[62]
	dedifferentiated liposarcoma	decreased	-	-	[66]

CSF1R—colony-stimulating factor receptor 1, SIRPA—signal regulatory protein alpha.

Furthermore, in osteosarcoma, a subset of M1 macrophages (INOS+) with osteoprotegerin (OPG)+ macrophages was related to a non-metastatic process [2,74]. Opposite results were obtained in the preclinical models of osteosarcoma, in which CD163+ M2 macrophages were correlated with tumor growth, vascularization, and metastasis [75]. A high number of CD14+/HLA-DRα+ macrophages was also associated with a better OS, vascularity and a better response to chemotherapy treatment in osteosarcoma; however, the M2 phenotype of CD14+/CD163+ macrophages was not related to OS [58].

Due to the heterogeneity of sarcomas and the dual role of TAMs as prognostic factors, there is a huge need for further research in this field to better understand TAM phenotype and their mechanism of action among each sarcoma subtype.

## 5. Macrophages as Treatment Targets in Sarcoma

Due to the variety of functions of TAMs and their clinical significance, they have become a therapeutic target among STS and bone sarcomas. Currently, several clinical trials using monocytes and TAMs as the target in sarcomas are ongoing (Table 3). Targeted therapies using TAMs can be divided into those aimed at reducing macrophage recruitment, the repolarization of TAMs into antitumor, targeting activated TAMs, or causing the activation of phagocytosis [2,45].

TLR agonists have been used to assess macrophage polarization toward M1 activation status. One of these agents is muramyl tripeptide-phosphatidylethanolamine encapsulated in liposomes (L-MTP-PE, Mifamurtide, Mepact^®^, London, UK), which has been suggested as an adjuvant therapy for patients with osteosarcoma [2,39,45]. Mifamurtide stimulates the response of the immune system by binding to nucleotide-binding oligomerization domain-containing protein 2 (NOD2) in an intracellular receptor molecule, expressed in monocytes, macrophages, and dendritic cells [2]. L-MTP-PE has been shown to cause TLR4 activation in macrophages and monocytes. In addition, it can upregulate the pro-tumor functions of macrophages by increasing cytokine production (such as TNF-α, NO, IL-1, IL-6, IL-8, and IL-12) [45]. In phase II clinical trials, L-MTP-PE was found to cause the infiltration of activated macrophages into lung tissue in osteosarcoma metastases [76]. On the other hand, in a follow-up randomized phase III trial [Intergroup (INT)-0133] involving patients with osteosarcoma, the supplement of L-MTP-PE to standard chemotherapy did not demonstrate differences in the 5-year OS or event-free survival (EFS) [77]. However, L-MTP-PE improved the 5-year OS and EFS in patients with metastatic disease, compared to those who did not receive L-MTP-PE (53 vs. 40% OS and 46% vs. 26% EFS) [45,77]. Punzo et al. [78], showed that L-MTP-PE may modulate macrophage functions, with the inhibition of osteosarcoma cell proliferation by switching macrophage polarity to TAM with an intermediate M1/M2 phenotype. A preclinical evaluation of L-MTP-PE in combination with zoledronic acid (ZA) in osteosarcoma murine models also has shown that these two drugs significantly suppress tumor growth and the development of metastatic disease [79]. Other potential agents involved in TAM repolarization that were tested in preclinical models in OS models were all-trans-retinoic acid, asiaticoside, Resiquimod (R848), and esculetin [80].

PLX3397 (Turalio^®^; Daiichi Sankyo, Inc., Tokyo, Japan), which is a CSF1/CSF1R inhibitor and agent that changes macrophage polarization, was tested in sarcomas in numerous clinical trials (NCT01004861, NCT01525602, NCT02390752, and NCT02584647) [2]. Preclinical studies have demonstrated that the CSF1/CSF1R signaling cascade reduces the number of infiltrating TAMs and causes M2 to M1 repolarization [2,81]. PLX3397 has been approved by the U.S. Food and Drug Administration in the treatment of unresectable tenosynovial giant cell tumors, and, currently, together with sirolimus, is being examined in a phase I/II trial for unresectable sarcomas (such as ES, liposarcoma, LMS, MPNST, SS and rhabdomyosarcoma) (NCT02584647) [2,82].

Imatinib (STI571, Glivec^®^; Novartis AG, Basel, Switzerland) is a strong and selective inhibitor of several tyrosine kinases, including platelet-derived growth factor receptor (PDGFR), which is the therapeutic option for patients with GIST [83]. Cavnar et al. [84], indicated that TAMs in GISTs are similar to M1 macrophages at baseline; however, they become similar to M2 macrophages in response to the inhibition of the KIT (CD117) oncoprotein by imatinib treatment. As a result, KIT inhibition leads to tumor cell apoptosis, the down-regulation of M1 macrophages, and up-regulation of M2 markers, which then causes T cell inhibition and the dual consequence of this treatment [84]. TAMs have been reported to be M1-like in untreated tumors; however, they change to M2-like in response to imatinib and eventually return to M1-like in imatinib-resistant GIST [84,85].

Another multikinase inhibitor, regorafenib, can induce antitumor activity by modulating macrophage polarization. It was observed in in vivo and in vitro liver cancer models by activating the p38MAPK/cyclic adenosine monophosphate (cAMP) responsive element binding protein 1 (Creb1)/Krupple-like factor 4 (Klf4) pathway after regorafenib administration [86]. The randomized double-blind phase II study of the regorafenib in patients with metastatic osteosarcoma (NCT02048371) revealed that the median PFS was significantly improved with the regorafenib (3.6 months (95% CI, 2.0 to 7.6 months) versus 1.7 months (95% CI, 1.2 to 1.8 months) in the placebo group); however, there were no statistically significant differences in the OS of these groups [87].

Trabectedin (Yondelis^®^; PharmaMar, Madrid, Spain) is another potential agent in sarcoma treatment, which has been shown to activate a TNF-related apoptosis-inducing ligand (TRAIL) in TAMs and monocytes, and it is involved in the TRAIL/caspase 8-dependent pathway of apoptosis [88,89]. TRAIL expression has been indicated, e.g., in the human ES cell line or murine fibrosarcoma cells [89,90]. Trabectedin has recently been approved for the treatment of advanced STS, and it was reported to partially decrease circulating monocytes and TAMs in myxoid liposarcoma cells [91]. Furthermore, Germano et al. [91], also showed that trabectedin decreased the TAM density and reduced angiogenesis in human sarcoma cells.

Another strategy relies on blocking specific receptors in TAMs. One of these examples may be PD-1/PD-L1 signaling using, for example, pembrolizumab, a highly selective humanized monoclonal antibody. In a phase II study of four cohorts of patients with STS, including LMS, UPS, GIST, and others, patients received pembrolizumab combined with metronomic cyclophosphamide (NCT02406781) [60]. The results of the 6-month PFS rates were 0% for LMS and UPS, 11% for GIST, and 14% for others [60]. Furthermore, in the same study, IDO1 was expressed mainly by macrophages in, respectively, 69%, 73%, 63%, and 29% of LMS, UPS, GIST, and others. Therefore, PD-1 inhibition has decreased activity in some GIST and STS, which may be explained by macrophage infiltration and activation of the IDO1 pathway, leading to immunosuppressive TME [60]. Due to the negative correlation of PD-1 expression in TAMs with phagocytic capacity against tumor cells, macrophage-targeted therapy with immune checkpoint modification can be a target for directed therapies [2]. The preclinical study showed that the combination of a CSF1R inhibitor with PD-1 or CTLA-4 antagonists induces tumor regression, compared to the single use of PD-1 or CTLA-4 inhibitors. However, the phase I/IIa clinical trial of PLX3397 for patients with advanced melanoma and other solid tumors was not continued due to unsatisfactory clinical efficiency (NCT02452424) [2]. On the other hand, a study by Edris et al. [92] reported that the inhibition of CD47/SIRPA signaling may be another promising target in the future. In this study, CD47 inhibition increased the macrophage phagocytic capacity in two human LMS cell lines (LMS04 and LMS05) in vitro. Furthermore, it reduced the primary tumor size in a metastatic murine model of LMS [92].

## 6. Conclusions

TAMs are a specific group of immune cells in TME and have been widely indicated in many sarcomas, including both STSs (i.e., DFSP, EMC, LGFMS, MFS, MPNST, ASPS, epithelioid sarcoma, clear cell sarcoma, angiosarcoma, liposarcoma, SS, LMS, UPS, GIST, rhabdomyosarcoma, fibrosarcoma) and bone sarcomas (adult and pediatric osteosarcoma, chondrosarcoma, chordoma, ES). TAMs have a great impact on both the tumor and surrounding immune cells, such as T cells, NK cells, and other macrophages. TAMs participate in tumor migration and invasion, and lead to an immunosuppression mechanism in many subtypes of sarcomas, through the expression of several markers (PD-1, PD-L1, CD80, SIRPA, CD206, CD209, CD146, CD14, CD68, CD163, CSF1R, etc.) and the secretion of multiple cytokines (IL-10, IL-6, IL-8, CSF1, IL-34, TGF-β, VEGF, etc.), chemokines (CCl2, CCL5, CCL18, CCL22, etc.) and other factors (such as MMP-9, COX2, and IDO1 etc.). In this process, various important pathways, including the JAK/MAPK, JAK/STAT3, p-STAT3/p-Erk1/2, and JAK/PI3K/AKT signaling pathways seem to be involved.

In addition, TAMs can be significant prognostic and predictive factors. They are generally related to a poor OS and the appearance of metastases among sarcomas, for example, chondrosarcoma, osteosarcoma, liposarcoma, UPS, or LMS, especially TAMs with the expression of CD163, CD68, SIPRA, CD204, or CD206. On the other hand, in some sarcomas, e.g., ES or osteosarcoma, the infiltration of TAMs was a favorable prognostic or predictive factor. The variety of TAM function and their clinical significance have led to research on targeting TAMs, as potential therapeutics in the treatment of STS and bone sarcomas. Among them, therapies related to TAM repolarization, targeting activated TAMs, phagocytosis activation, and TLR antagonists are under development. Currently, Pexidartinib and Trabectedin are approved for sarcoma treatment; however, there are many ongoing clinical trials with other drug candidates.

In conclusion, there is still much to be discovered about TAMs and their role in the pathogenesis of different types of sarcomas and their impact on TME. Their impact on clinical outcomes is still unclear, which may be due to the heterogeneity of these tumors and their rare occurrence, making them a difficult target for research. However, TAMs appear to be a promising target for treatment due to their wide distribution among many subtypes of sarcoma and the presence of specific markers on their surface. Further studies are needed to better understand the mechanism of the repolarization of TAMs and their differential impact on clinical significance among sarcomas, which may lead to the development of effective targeted therapies using TAMs in the future.

## Figures and Tables

**Figure 1 cancers-15-05294-f001:**
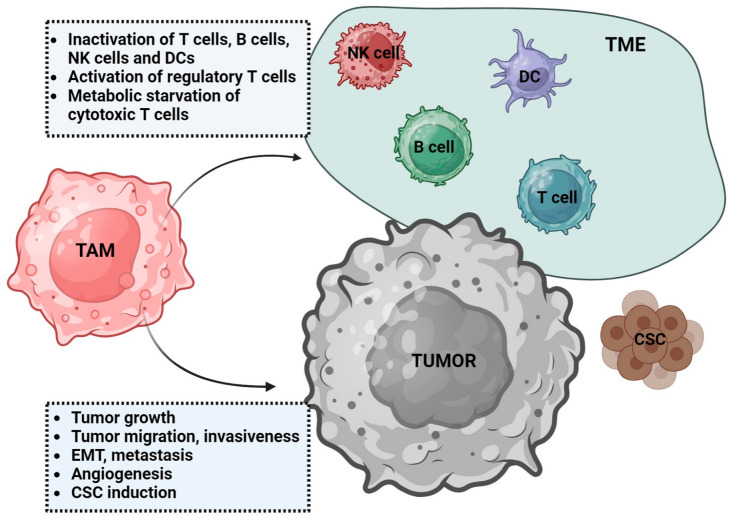
Dual role of tumor-associated macrophages (TAMs). TAMs influence tumor microenvironment (TME) leading to immunosuppression. In addition to modulating immune cells, they promote tumor growth via induction of cancer stem cells (CSC), angiogenesis, or epithelial-mesenchymal transition (EMT). DCs—dendritic cells, NK cells—natural killer cells.

**Figure 2 cancers-15-05294-f002:**
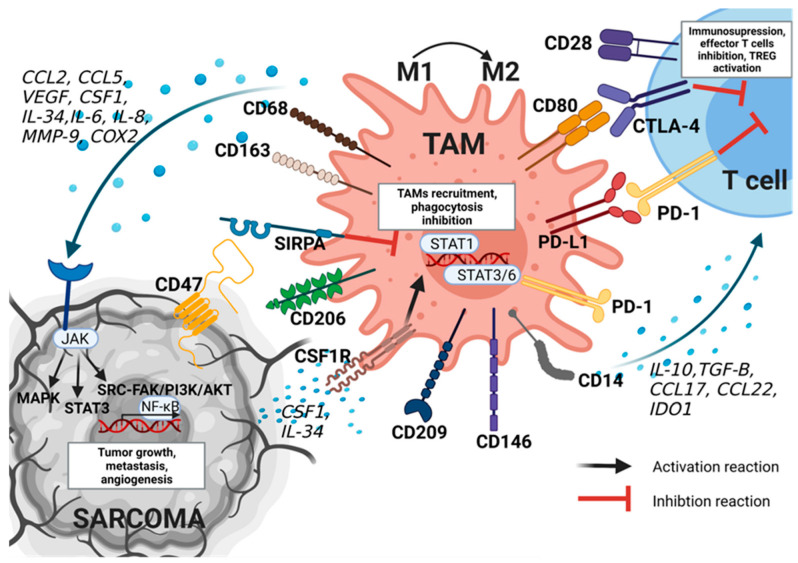
Tumor-associated macrophages (TAMs) and their impact on the microenvironment of sarcomas. TAMs display a pro-tumor phenotype with polarization toward M2-type macrophages. TAMs lead to modulation of tumor microenvironment (TME) by CD80/CTLA-4 and PD-1/PD-L1 interaction, secretion of MMP-9, COX2, IDO1, and several cytokines and chemokines e.g., IL-10, Il-34, IL-6, TGF-B, CSF1, VEGF. All of these factors lead to immunosuppression by activating regulatory T cells (TREG) and inactivating effector T cells. On the other hand, by activating the JAK/MAPK, JAK/STAT3, and JAK/PI3K/AKT signaling pathways, TAMs cause activation of NF-κB transcription factor and, as a result, tumor progression and promotion. Sarcoma cells, through the secretion of cytokines and the CSF1R interaction, lead to the recruitment of TAMs. Furthermore, through CD47/SIRPA signaling, tumor cells inhibit the function of phagocytosis in TAMs. AKT—Protein Kinase B; CCL—C-C motif chemokine ligand; COX2—cyclooxygenase-2; CSF1—colony-stimulating factor 1; CSF1R—colony-stimulating factor 1 receptor; CTLA-4—Cytotoxic T-Lymphocyte Associated Protein 4; FAK—Focal adhesion kinase; IDO1—indoleamine 2,3-dioxygenase 1; IL- interleukins; JAK—Janus-Activated Kinase 2; MAPK—mitogen-activated protein kinase; MMP-9—matrix metalloproteinase-9; NF-κβ—nuclear factor-κB; PD-1—programmed death cell receptor; PD-L1—programmed death-ligand 1; PI3K—phosphatidylinositol 3-kinase; SIRPA—signal regulatory protein alpha; SRC—proto-oncogene tyrosine-protein kinase; STAT—signal transducer and activator of transcription; TGF-B—transforming growth factor β; VEGF—vascular epithelial growth factors.

**Table 3 cancers-15-05294-t003:** Ongoing clinical trials using macrophages as a target for immunotherapy among sarcomas. Clinical trials have been found using the following keywords: sarcomas, macrophages; solid tumors, macrophages (https://clinicaltrials.gov and https://www.clinicaltrialsregister.eu/, accessed on 1 September 2023).

Clinical Trial Number	Status	Conditions	Interventions	Phases
NCT02584647	Recruiting	Sarcoma and malignant peripheral nerve sheath tumors	Drug: PLX3397 (Receptor Tyrosine Kinase Inhibitor), Sirolimus	Phase I/II
NCT02502786	Recruiting	Recurrent osteosarcoma	Biological: Humanized anti-GD2 antibody Drug: GM-CSF	Phase II
NCT03217266	Active, not recruiting	Soft tissue sarcoma	Drug: MDM2 Inhibitor KRT-232 Radiation: Radiation Therapy	Phase I
NCT01050296	Recruiting	Pediatric solid tumors	Not applicable	Not applicable
NCT04465643	Recruiting	Nerve sheath tumors	Drug: Nivolumab, Ipilimumab	Phase I
NCT03866525	Recruiting	Solid tumor, gastrointestinal cancer	Biological: OH2 (anti-PD-1 antibody) injection, with or without irinotecan or HX008	Phase I/II
NCT02389244	Recruiting	Ewing sarcomas, chondrosarcomas, osteosarcomas, chondroma, CIC-rearranged sarcoma	Drug: Regorafenib, placebo	Phase II
NCT05427461	Recruiting	Leiomyosarcoma	Other: Blood samples will be collected at different times	Not applicable
NCT02390752	Recruiting	Neurofibroma, plexiform, precursor cell, lymphoblastic leukemia-lymphoma, leukemia, promyelocytic, acute, sarcoma	Drug: Turalio (CSF1R inhibitor)	Phase I
NCT04900519	Recruiting	Solid tumor relapsed solid neoplasm refractory tumor	Biological: STI-6643 Anti-CD47 human monoclonal antibody	Phase I
NCT03990233	Recruiting	Solid tumor, adult	Drug: BI 765063 (antagonist to SIRPα receptor), BI 754091 (antagonist to PD-1 receptor)	Phase I
NCT04660929	Recruiting	All HER2 overexpressing solid tumors	Biological: CT-0508, Pembrolizumab	Phase I
NCT04168528	Recruiting	Malignant solid tumor, breast cancer, head and neck cancer, melanoma	Drug: 68GaNOTA-Anti-MMR-VHH2	Phase I/II

## Data Availability

The data can be shared up on request.

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
