# Peer review of "The Role of Macrophages in Sarcoma Tumor Microenvironment and Treatment"

_cancers, 2023, doi:10.3390/cancers15215294_

Round 1

Reviewer 1 Report

Comments and Suggestions for Authors

This review addresses the role of TAMs in STS and bone Sarcoma TME, reporting the specific markers and cell interactions involved in tumour development among different types and subtypes of Sarcomas. Authors also describe the clinical significance of macrophages and their role as future therapeutic targets.

The authors did a complex work, though helped by many Reviews and articles on Macrophages and their functions in cancer recently reported.  Similarly complex and rather innovative is the work of describing the impact of macrophages in Sarcomas.

Overall, the Review deals with the important and indispensable problems regarding macrophages in cancer and specifically in Sarcomas. However, in order to increase the impact of this Review some changes are suggested.

Concerns

Line 63, “phagocytose” , phagocytize would be correct.

In each paragraph of the review the authors report such extensive and detailed bibliographical documentation that often the final meaning for which the paragraph was written is lost.

An example is the paragraph of the “Macrophage Polarization” where many acquisitions of literature are reported, but in the end the intimate meaning of polarization or activation are vague. Authors should provide a conclusion to the paragraph in order to correlate and interpret, where possible, existing and reported scientific data. In this way, the various topics dealt with in the paragraphs will no longer appear as a mere list of published scientific results, but rather will be the subject of discussion and open issues. In this paragraph the mechanism of polarization of macrophages is invariably used as activation. Perhaps it is good to clarify whether polarization and activation are the same process or different mechanisms that characterize different  functional phenotypes, M1 or M2 for example.

Line 163-167, the sentence “In TME, TAMs induce………………and metalloproteinases (MMPs) et al. [26, 27].” The suggestion is to better clarify the concept even by spending a few more words in favour of clarity not only bibliography that the readers must read to understand.

Line 248. Authors are invited to clarify the following sentence: “an antiphagocytic signal that separates living cells from dying cells”.  Still, authors could spend a few extra words to improve understanding.

The paragraph "Distribution of TAMs in Sarcoma" is a list of bibliographical data that report the preponderant presence of macrophages in patients suffering from various types of sarcoma (many!). Macrophages with specific membrane markers are also reported for various types of sarcoma. The predictive and prognostic value of macrophages in sarcomas was reported in the next paragraph without addressing the problem of their opposite role of positive or negative prognosis, this  aspects  deserve more attention,  especially in light of a therapy targeting monocyte/macrophage.

Conclusions are the part in which the authors can put forward working hypotheses and discuss the still unresolved problems. The conclusions appear hasty and insufficient to introduce the problem of targeting macrophages as possible therapy in Sarcomas, problem somehow introduced by the title.

The Review is full of Acronyms,  it would be better to make a list of them with their meanings.

Author Response

Response to Reviewer 1

Thank you very much for taking the time to review this manuscript. Please find the detailed responses below and the corresponding corrections in track changes in the re-submitted file.

"This review addresses the role of TAMs in STS and bone Sarcoma TME, reporting the specific markers and cell interactions involved in tumour development among different types and subtypes of Sarcomas. Authors also describe the clinical significance of macrophages and their role as future therapeutic targets.

The authors did a complex work, though helped by many Reviews and articles on Macrophages and their functions in cancer recently reported.  Similarly complex and rather innovative is the work of describing the impact of macrophages in Sarcomas.

Overall, the Review deals with the important and indispensable problems regarding macrophages in cancer and specifically in Sarcomas. However, in order to increase the impact of this Review some changes are suggested."

Thank you for all your valuable comments and suggestions, below we present our responses.

Concerns

  1. Line 63, “phagocytose” , phagocytizewould be correct.

Response 1: Thank you for this comment, it has been corrected.

  1. In each paragraph of the review the authors report such extensive and detailed bibliographical documentation that often the final meaning for which the paragraph was written is lost.

An example is the paragraph of the “Macrophage Polarization” where many acquisitions of literature are reported, but in the end the intimate meaning of polarization or activation are vague. Authors should provide a conclusion to the paragraph in order to correlate and interpret, where possible, existing and reported scientific data. In this way, the various topics dealt with in the paragraphs will no longer appear as a mere list of published scientific results, but rather will be the subject of discussion and open issues. In this paragraph the mechanism of polarization of macrophages is invariably used as activation. Perhaps it is good to clarify whether polarization and activation are the same process or different mechanisms that characterize different  functional phenotypes, M1 or M2 for example.

Response 2: Thank you for this comment. We aimed to introduce the mechanism of macrophages polarization to better understand the TAMs characterization and presence of both M1 and M2 markers. However, we added an introduction and conclusion sentence to this paragraph as follow to make it more clearly:

“Based on different stimuli macrophages can polarize into M1 or M2 macrophages with different activation ways. Classical macrophage activation, which occurs in M1 can occur as ais a result of cell stimulation by many factors, including 1) interferon-gamma (IFN-γ), secreted primarily by Th1 helper CD4+ T cells, cytotoxic CD8+ T cells, and natural killer cells (NK); 2) lipoproteins and lipopolysaccharide………… Moreover,  response gene to complement 32 (RGC-32), which is a cell cycle regulator, play also an important role in M2 polarization [8].

The factors presented influence the phenotype of macrophages, which varies depending on the stimuli they receive. Some of these factors are present in TME and are involved in the development of specific macrophages that participate, e.g., in tumor growth.”

  1. Line 163-167, the sentence “In TME, TAMs induce………………and metalloproteinases (MMPs) et [26, 27].” The suggestion is to better clarify the concept even by spending a few more words in favour of clarity not only bibliography that the readers must read to understand.

Response 3: Thank you for this comment. We paraphrase this sentence to make more clear its meaning:

“TAMs influence both TME and tumor cells (Figure 1). In TME, TAMs secrete various cytokines, chemokines and enzymes, such as e.g. TGF-β, prostaglandins, IL-10, CCL22, CCL17, galectin-3 (Gal-3) and metalloproteinases (MMP) that induce immunosuppression by activation of regulatory T cells (TREG), which in consequence lead to loss of T cell function to recognize and kill tumor cells [1,26,27].

  1. Line 248. Authors are invited to clarify the following sentence: “an antiphagocytic signal that separates living cells from dying cells”.  Still, authors could spend a few extra words to improve understanding.

Response 4: Thank you for this comment, to clarify we change this sentence as follow:

“SIRPA recognizes CD47, which negatively regulates the effector function of innate immunity cells, such as phagocytosis of the host, and participates in homeostasis processes by removing dying cells”

  1. The paragraph "Distribution of TAMs in Sarcoma" is a list of bibliographical data that report the preponderant presence of macrophages in patients suffering from various types of sarcoma (many!). Macrophages with specific membrane markers are also reported for various types of sarcoma. The predictive and prognostic value of macrophages in sarcomas was reported in the next paragraph without addressing the problem of their opposite role of positive or negative prognosis, this  aspects  deserve more attention,  especially in light of a therapy targeting monocyte/macrophage.

Response 5: Thank you for this comment. That is true that in this paragraph is a lot of data and it can be confusing, however this group of tumors is very heterogenous (>100 subtypes) and there is not ordered data for them in this field. We wanted to present here a comprehensive collection of results known so far to present the potential targets for future therapies by describing discovered markers among sarcomas. Not much is known about the mechanism why in some sarcomas TAMs have positive or prognostic role, so we showed the results of both negative and positive role in different paragraph in section about predictive and prognostic factors in sarcoma and summarized them also in table 2 for better understanding so many data and to show that there is a need to conduct further researches.

We added a following conclusions to this two paragraphs to better address the problem as follow:

“The distribution of TAMs among sarcomas is very heterogeneous, as a whole group of these tumors. For this reason, there is still not much known about the specific markers of TAMs and their interaction in sarcomas. This may impact the clinical relevance of TAMs and their conflicting function on clinical outcomes among different subtypes of sarcomas.”

And

“Due to the heterogeneity of sarcomas and the dual role of TAMs as prognostic factors, there is a huge need for further research in this field to better understand TAMs phenotype and their mechanism of action among each sarcoma subtype.”

  1. Conclusions are the part in which the authors can put forward working hypotheses and discuss the still unresolved problems. The conclusions appear hasty and insufficient to introduce the problem of targeting macrophages as possible therapy in Sarcomas, problem somehow introduced by the title.

Response 6: Thank you for this suggestion. We were trying to make a summary of whole article to clarify this complex issue and also make some hypothesis about potential pathways involving in immunosuppression. We outlined problem of heterogeneity and contrary role of TAMs in sarcomas which is an introduction for further research. However, according to your suggestion to focus more on problem of targeting macrophages as possible therapy we modify the paragraph as follow:

“In conclusion, there is still much to be discovered about TAMs and their role in the pathogenesis of different types of sarcomas and their impact on TME. Their impact on clinical outcomes is still unclear, which may be due to the heterogeneity of these tumors and their rare occurrence, making them a difficult target for research. However, TAMs appear to be a promising target for treatment due to their wide distribution among many subtypes of sarcoma and the presence of specific markers on their surface. Further studies are needed to better understand the mechanism of repolarization of TAMs and their differential impact on clinical significance among sarcomas, which may lead to the development of effective targeted therapies using TAMs in the future.”

  1. The Review is full of Acronyms,  it would be better to make a list of them with their meanings.

Response 7: Thank you for this suggestion we added the list of all abbreviations at the end of the manuscript.

Reviewer 2 Report

Comments and Suggestions for Authors

This manuscript deals with tumor associated macrophages (TAM) in sarcomas. The authors describe in detail the features of the different subsets of TAM and the role in the tumor microenvironment (TME). Also, some suggestions on the therapy focusing on targeting of TAM.

Overall, the manuscript is well organized and written.

I would suggest to insert and discuss some additional reports that follow.

CSF1/CSF1R Signaling Inhibitor Pexidartinib (PLX3397) Reprograms Tumor-Associated Macrophages and Stimulates T-cell Infiltration in the Sarcoma Microenvironment. Fujiwara T, Yakoub MA, Chandler A, Christ AB, Yang G, Ouerfelli O, Rajasekhar VK, Yoshida A, Kondo H, Hata T, Tazawa H, Dogan Y, Moore MAS, Fujiwara T, Ozaki T, Purdue E, Healey JH.Mol Cancer Ther. 2021 Aug;20(8):1388-1399. doi: 10.1158/1535-7163.MCT-20-0591. Epub 2021 Jun 4. PMID: 34088832
Association of SMC4 with prognosis and immune infiltration of sarcoma.Jiang G, Chen J, Li Y, Zhou J, Wang W, Wu G, Zhang Y. Aging (Albany NY). 2023 Jan 30;15(2):567-582. doi: 10.18632/aging.204503. Epub 2023 Jan 30.PMID: 36719264

Macrophage Repolarization as a Therapeutic Strategy for Osteosarcoma.Anand N, Peh KH, Kolesar JM.Int J Mol Sci. 2023 Feb 2;24(3):2858. doi: 10.3390/ijms24032858. PMID: 36769180

The proteomic landscape of soft tissue sarcomas.Burns J, Wilding CP, Krasny L, Zhu X, Chadha M, Tam YB, Ps H, Mahalingam AH, Lee ATJ, Arthur A, Guljar N, Perkins E, Pankova V, Jenks A, Djabatey V, Szecsei C, McCarthy F, Ragulan C, Milighetti M, Roumeliotis TI, Crosier S, Finetti M, Choudhary JS, Judson I, Fisher C, Schuster EF, Sadanandam A, Chen TW, Williamson D, Thway K, Jones RL, Cheang MCU, Huang PH. Nat Commun. 2023 Jun 29;14(1):3834. doi: 10.1038/s41467-023-39486-2. PMID: 37386008

Comments on the Quality of English Language

The English language is good.

Author Response

Response to Reviewer 2

Thank you very much for taking the time to review this manuscript. Please find the detailed responses below and the corresponding corrections in track changes in the re-submitted file.

This manuscript deals with tumor associated macrophages (TAM) in sarcomas. The authors describe in detail the features of the different subsets of TAM and the role in the tumor microenvironment (TME). Also, some suggestions on the therapy focusing on targeting of TAM.

Overall, the manuscript is well organized and written.

Thank you for all your valuable comments and suggestions, below we present our responses.

I would suggest to insert and discuss some additional reports that follow.

1. CSF1/CSF1R Signaling Inhibitor Pexidartinib (PLX3397) Reprograms Tumor-Associated Macrophages and Stimulates T-cell Infiltration in the Sarcoma Microenvironment. Fujiwara T, Yakoub MA, Chandler A, Christ AB, Yang G, Ouerfelli O, Rajasekhar VK, Yoshida A, Kondo H, Hata T, Tazawa H, Dogan Y, Moore MAS, Fujiwara T, Ozaki T, Purdue E, Healey JH.Mol Cancer Ther. 2021 Aug;20(8):1388-1399. doi: 10.1158/1535-7163.MCT-20-0591. Epub 2021 Jun 4. PMID: 34088832

Response 1: Thank you for this suggestion it was added as follow:

“In a study by Fujiwara et al. in vitro administration of Pexidartinib (PLX3397) reduced TAMs M2-like polarization to M1-like phenotype with a simultaneous decrease in fork-head box P3 (FOXP3)+ regulatory T cells and increased migration of CD8+ T cells [40]. Moreover, in the same study, in osteosarcoma mice model PLX3397 administration suppressed the primary tumor growth and lung metastasis in vivo [40].”

2. Association of SMC4 with prognosis and immune infiltration of sarcoma.Jiang G, Chen J, Li Y, Zhou J, Wang W, Wu G, Zhang Y. Aging (Albany NY). 2023 Jan 30;15(2):567-582. doi: 10.18632/aging.204503. Epub 2023 Jan 30.PMID: 36719264

Response 2: Thank you for this suggestion it was added as follow:

“Moreover, in a study by Jiang et al. the expression of structural maintenance of chromo-somes 4 (SMC4) was significantly correlated with CD14+ and CSF1R+ monocyte, CD80+ TAMs, prostaglandin-endoperoxide synthase 2 (PTGS2)+ M1, and expression of de-pleted CTLA4+ T cells [53]. This suggests that SMC4 may play a role in regulating the infiltration and activity of TAM and T cells function inhibition and, therefore may promote the development of sarcoma [53].”

3. Macrophage Repolarization as a Therapeutic Strategy for Osteosarcoma.Anand N, Peh KH, Kolesar JM.Int J Mol Sci. 2023 Feb 2;24(3):2858. doi: 10.3390/ijms24032858. PMID: 36769180

Response 3: Thank you for this suggestion it was added as follow:

“Other potential agents involved in TAMs repolarization that were tested in preclinical models in OS models were all-trans-retinoic acid, asiaticoside, Resiquimod (R848), and esculetin [80].”

4. The proteomic landscape of soft tissue sarcomas.Burns J, Wilding CP, Krasny L, Zhu X, Chadha M, Tam YB, Ps H, Mahalingam AH, Lee ATJ, Arthur A, Guljar N, Perkins E, Pankova V, Jenks A, Djabatey V, Szecsei C, McCarthy F, Ragulan C, Milighetti M, Roumeliotis TI, Crosier S, Finetti M, Choudhary JS, Judson I, Fisher C, Schuster EF, Sadanandam A, Chen TW, Williamson D, Thway K, Jones RL, Cheang MCU, Huang PH. Nat Commun. 2023 Jun 29;14(1):3834. doi: 10.1038/s41467-023-39486-2. PMID: 37386008

Response 4: Thank you for this suggestion, the fragment about TAMs markers were added as follow:

In a study by Burns et al., a poor correlation between CD163 and STAT6 was found in dedifferentiated liposarcoma and in undifferentiated pleomorphic sarcoma (UPS) [52].”